# A Century of Clinical Use of Phages: A Literature Review

**DOI:** 10.3390/antibiotics12040751

**Published:** 2023-04-13

**Authors:** Kevin Diallo, Alain Dublanchet

**Affiliations:** 1Department of Infective and Tropical Diseases and Internal Medicine, University Hospital of la Reunion, 97448 Saint-Pierre, France; 2Independent Researcher, 2465 Rue Céline Robert, 94300 Vincennes, France

**Keywords:** phage therapy, bacteriophage, clinical use

## Abstract

Growing antibiotic resistance and the broken antibiotic market have renewed interest in the use of phages, a century-old therapy that fell into oblivion in the West after two decades of promising results. This literature review with a particular focus on French literature aims to complement current scientific databases with medical and non-medical publications on the clinical use of phages. While several cases of successful treatment with phages have been reported, prospective randomized clinical trials are needed to confirm the efficacy of this therapy.

## 1. Introduction

Growing antibiotic resistance and the broken antibiotic market currently pose a major threat to global health. In this context, new therapeutic strategies are needed to avoid returning to the pre-antibiotic era. One potential solution is the use of phages, a century-old therapy that fell into oblivion in the West after two decades of promising results. Phages (or bacteriophages) are bacteria-specific viruses that inject their genome into bacterial cells and use the bacterial metabolism to replicate. The life cycle of phages is either lytic or lysogenic. During the lytic cycle, the injected phage genome replicates and destroys the bacterial cell, thus inhibiting bacterial growth. During the lysogenic cycle, the injected genome remains in the host’s genome, where it enters a dormant state [1,2]. Lytic phages will be the focus of this literature review, as only they have the capacity to destroy infected bacteria.

Phages are well absorbed enterally and transmucosally [3]. They have short half-lives in vivo as they are rapidly destroyed by the immune system, in particular the reticulo-endothelial system. When they infect their target bacteria, however, they can multiply exponentially [4]. Phages have been classified by the International Committee for Taxonomy of Viruses into 12 families, with *Straboviridae*, *Autographiviridae,* and *Drexlerviridae* accounting for over 95% of identified phages [5]. Phages present several advantages: they self-replicate; they have greater specificity than antibiotics; they can be used in patients with allergy to antibiotics; they have a low rate of side effects; their production cost is low; they can be administered through different routes; they can present synergistic effects when combined with antibiotics; they are effective against biofilm; and in the case of bacterial resistance to a specific phage, other phages can be used against the resistant bacterium [1,6]. However, phages also have disadvantages: phage treatment can be initiated only after the causative agent has been identified; multi-bacterial infections require cocktails containing several phages; phage preparations must be sterilized; phage preparations have to be very clean and endotoxin free; phage treatment can induce an immune response in humans; pharmacokinetic data on the action of phages in the human body are lacking; and phages have yet to be properly classified by regulatory agencies [1,6,7].

Many cases of successful treatment with phages have been reported over the last century. Yet, several of these reports are missing from current scientific databases: some fell into oblivion before the advent of databases, others were published in non-medical outlets, and others were written in languages other than English [8]. While many case reports of treatment with phages have an insufficient level of proof, a century of experimentation cannot be ignored.

This literature review with a particular focus on French literature aims to complement current scientific databases with medical and non-medical publications on the clinical use of phages in humans.

## 2. Context

### 2.1. History of Phage Therapy

Phages were first described as ultra-microscopic viruses by Frederick Twort in 1915 and characterized as bacteriophages by Félix d’Hérelle two years later [9,10]. They were introduced in the former Eastern Bloc by a student of d’Hérelle, George Eliava, who founded a research institute specifically dedicated to phage therapy in Tbilissi, Georgia, in 1923.

In the 1920s and 1930s, phages were used worldwide for several indications in humans. Several large pharmaceutical companies produced commercial phage preparations at the time: Eli Lilly (Indianapolis, IN, USA) in the early 1930s, Abbott Labs (Chicago, IL, USA) in the early 1930s, and Bristol-Myers Squibb (New York, NY, USA) from the early 1930s to the 1940s [11]. However, phage therapy failed to impose itself in the West due to inconsistent treatment results, a poor understanding of phages’ mechanism of action, and the advent of broad-spectrum antibiotics [1]. The therapy was discredited in two reports published in 1934 and 1941 by Eaton and Krueger [12,13,14,15,16]. While some Western companies continued to produce phages afterwards (in particular for the food processing industry), the use of phages in humans was largely abandoned. 

In the former Eastern Bloc, the institute founded by Eliava in Georgia continued to carry out major research on phage therapy and supplied phage products to the USSR throughout the 20th century. Renamed the G. Eliava Institute of Bacteriophages, Microbiology, and Virology in 1988, it pursued its research activities after the fall of the Iron Curtain. In Russia, phages are now produced commercially by the company Microgen.

In 2005, a Phage Therapy Unit was created at the Hirszfeld Institute of Immunology and Experimental Therapy in Wrocław, Poland. This unit is now the second largest center for phage research in Europe after the G. Eliava Institute.

Since the mid-2000s, growing antibiotic resistance has prompted institutes and pharmaceutical laboratories all over the world to create their own phage banks on the model of the G. Eliava Institute [17]. These include: Adaptive Phage Therapeutics in the United States, Biobank, First Affiliated Hospital of Xi’an Jiaotong University, Institute for Protein Science and Phage Research in China, the Bacteriophage Bank of Korea [18], the Israeli Phage Bank at the Hebrew University of Jerusalem in Israel [19], the National Collection of Type Cultures, the Bacteriophage Collection in the United Kingdom, Fagenbank in the Netherlands, DMZ in Germany, Queen Astrid Military Hospital in Belgium, and Pherecydes Pharma in France.

Despite this renewed interest in phage therapy, the efficacy of phages has yet to be demonstrated.

### 2.2. Legal Framework for the Use of Phage Therapy in Humans

Phages are classified as drugs in the US and as medicinal products in the EU [20]. Like other drugs, they are subject to marketing and manufacturing authorization from the Food and Drug Administration (FDA) in the US and the European Medicines Agency (EMA) in the EU. 

To obtain marketing authorization, phage therapies must first be validated in preclinical in vitro and in vivo studies. Once these are completed, phase I to IV clinical trials are required to confirm their safety and efficacy in humans. To date, no phage therapy has reached phase IV of clinical trials.

To receive manufacturing authorization, phage products must comply with good manufacturing practice (GMP) standards, which involve a high level of purification and sterilization. However, given the lack of fit between these standards and the viral nature of phages, no phage product has so far been approved for use in humans by the EMA or the FDA. Phage production is authorized only for compassionate use and clinical research under Article 37 of the Declaration of Helsinki.

Marketing regulations and GMP standards are barriers limiting pharmaceutical investment in phage therapy [20].

### 2.3. Safety of Phage Therapy in Humans

The human body is routinely exposed to large numbers of endogenous phages. Since phages are composed only of proteins and DNA and do not release toxins when they die, their toxicity is low in humans. Phages must nevertheless be purified to reduce their virulence and ensure their safe administration [3].

Bacteria destroyed by phages can release bacterial toxins, and excessive bacterial lysis often causes immune system reactions. These phenomena, however, occur mainly when phages are administered intravenously. Other routes of administration can be used safely.

In 2009, Merabishvili et al. proposed a quality control for the safe use of purified phage cocktails against *Staphylococcus aureus* and *Pseudomonas aeruginosa* [21]. In 2014, an international panel composed of 29 experts from ten countries developed quality, safety, and efficacy requirements for sustainable phage therapy products [22]. Such requirements could replace current GMP standards, which would facilitate the use of phage therapy in humans.

### 2.4. Use of Phages in Food Processing and Plant Disease Control

Several studies have shown the efficacy and safety of using phage products in food processing and plant disease control [23,24,25,26].

Intralytix (Columbia, MA, USA) markets three phage products for food processing: ListShield™ against *Listeria monocytogenes*, EcoShield™ against *Escherichia coli*, and SalmoFresh™ against *Salmonella* spp. Omnilytics (Sandy, UT, USA) sells several phage products, including Agriphage™, for use in agriculture. Elanco (Greenfield, IN, USA) markets Finalyse™, an anti-bacterial spray targeting *E. coli* O157. In Europe, Micreos (Wageningen, The Netherlands) produces Listex™ and Salmonelex™, both of which were approved by the FDA. APS Biocontrol (Dundee, SC, UK) produces Biolyse^®^, a phage product sprayed on potatoes during processing. All these products have been approved by the FDA and/or the EMA [6].

## 3. Literature Review of a Century of Clinical Use of Phages

A search of the MEDLINE database was performed without language restrictions to identify articles on phage therapy published between 1922 and 2022. The following keywords were used: (phage OR phage therapy OR bacteriophage) AND (treatment OR use OR therapy) AND (clinical OR human). The titles and abstracts of identified articles were screened for inclusion. Prospective randomized clinical trials, literature reviews, prospective non-randomized clinical trials, and isolated case reports evaluating the clinical use of phages were included in the review. We excluded articles that did not deal with the use of phages in humans and those referring to articles already cited that did not bring new cases to our analysis.

The reference lists of selected articles were perused to identify other relevant publications on phage therapy. These publications consisted of books, theses, and medical and non-medical articles published between 1915 and 2022. Publications in languages other than English were considered, with a particular focus on French literature. All identified publications describing cases of phage therapy were included in the review.

The main prospective randomized clinical trials, literature reviews, and prospective non-randomized clinical trials are presented in Table 1.

Isolated case reports published after 1945 are shown in Table 2, while those published prior to 1945 appear in Table 3. We took 1945 as a cut-off year as this corresponds to the period when broad-spectrum antibiotic therapy was introduced and phage therapy lost credibility in the West. Moreover, phages were mostly used in isolation before 1945 and were generally combined with antibiotics after that date.

Studies were considered to have demonstrated the efficacy of phage treatment if 50% of evaluated patients had a favorable outcome. Studies that did not specify the number of improvements or cures were classified as unspecified. Figure 1 shows the distribution of articles on phage therapy for the main clinical foci of infection.

### 3.1. Prospective Randomized Clinical Trials 

Our search of the MEDLINE database identified nine prospective randomized trials evaluating phage therapy.

In 2021, Leitner et al. performed a randomized placebo-controlled clinical trial at the Alexander Tsulukidze National Centre of Urology in Tbilisi, Georgia, to assess the efficacy of Pyophage in 97 men with urinary tract infection [30]. Between 2017 and 2018, 28 men received Pyophage, 32 a placebo, and 37 an antibiotic treatment. No difference in treatment success rates was observed between the three groups.

In 2019, Febvre et al. and Gingin et al. presented the results of the Bacteriophage for Gastrointestinal Health (PHAGE) study, which aimed to determine the safety and tolerability of phages in healthy adults with mild to moderate gastrointestinal distress [39,40]. The PHAGE study was a randomized, double-blind, placebo-controlled crossover intervention, in which 32 patients received a treatment containing four strains of phages (LH01-Myoviridae, LL5-Siphoviridae, T4D-Myoviridae and LL12-Myoviridae) and a placebo, each for a period of 28 days. Febvre et al. reported that phage consumption caused minimal disruption to the gut microbiota. Gindin et al. found no effect of treatment sequence on comprehensive metabolic panel outcomes. 

In 2019, Ooi et al. published the results of a phase I clinical trial assessing the safety, tolerability, and preliminary efficacy of intranasal doses of experimental phage cocktail AB-SA01 in patients with recalcitrant chronic rhinosinusitis who had positive *S. aureus* cultures sensitive to AB-SA01 [46]. Three cohorts of three patients each received successive intranasal doses of AB-SA01 twice daily at a concentration of 3 × 10⁸ PFU for seven days (cohort 1), 3 × 10^8^ PFU for 14 days (cohort 2), and 3 × 10^9^ PFU for 14 days (cohort 3). Treatment was well tolerated overall, and no serious adverse events or deaths were reported. Preliminary efficacy results showed favorable outcomes, with clinical and microbiological evidence of infection eradication in two of the nine evaluated patients.

The results of the “PhagoBurn” phase I/II randomized clinical trial were published in 2018 by Jault et al. [52]. The aim of this trial was to compare the efficacy and tolerability of a cocktail of anti-*P. aeruginosa* phages to standard of care (silver sulfadiazine) in the treatment of burn wound infections. Twenty-seven patients were randomly allocated to receive a topical application of either treatment daily for 7 days, with 14 days of follow-up. Safety was evaluated in all patients who received at least one phage dressing (treatment group) or one silver sulfadiazine dressing (control group). However, in this study, the applied phage titer was extremely low, which contributed to clinical failure. The trial was stopped before termination due to insufficient efficacy of their phage cocktail.

The 2016 trial by Sarker et al., supported by a grant from Nestlé Nutrition and Nestlé Health Science, assessed the safety and efficacy of a T4-like phage cocktail compared to the Microgen ColiProteus phage cocktail and a placebo in Bangladeshi children hospitalized with acute bacterial infection or diarrhea [52]. No adverse event attributable to the oral application of phages was reported. Treated children had higher fecal phage prevalence and titers than those who received a placebo. However, the oral phages failed to achieve intestinal replication and to improve diarrhea outcome, possibly because phage coverage was insufficient and *E. coli* pathogen titers were too low.

In 2009, Rhoads et al. conducted a phase I trial to evaluate the safety of a phage cocktail for the treatment of venous leg ulcers in humans [59]. Forty-two patients with chronic venous leg ulcers were included, 39 of whom completed the trial. The ulcers were treated with either a saline control or a phage preparation targeting *P. aeruginosa*, *S. aureus*, and *E. coli* for a period of 12 weeks. Follow-up continued through week 24. No adverse events attributable to the phage cocktail were observed. No significant difference was found between the test and control groups in terms of frequency of adverse events, rate of cure or frequency of cure.

In 2009, Wright et al. published the results of a phase I/II trial assessing the safety and efficacy of the phage preparation Biophage-PA in patients with chronic otitis caused by an antibiotic-resistant strain of *P. aeruginosa* [60]. A total of 24 patients randomized into two groups of 12 were treated with a single dose of Biophage-PA or a placebo and were followed up 7, 21, and 42 days after local application by the same otologist. The main outcomes were the symptoms observed by physicians (erythema, inflammation, ulceration, granulation, polyps, amount of discharge, type of discharge, and odor) and those reported by patients (discomfort, itchiness, wetness, and smell). No adverse events were observed. In the phage-treated group, pooled clinical indicators significantly improved, and *P. aeruginosa* counts significantly decreased from baseline, demonstrating the efficacy of the phage preparation.

In 2005, Bruttin et al. evaluated the safety of *E. coli* phage T4 in 15 healthy adult volunteers [62]. All patients included in this crossover study received a low dose of phage T4 (103 plaque-forming units (PFU)/mL), a high dose of phage (10⁵ PFU/mL), and a placebo in their drinking water. No adverse event related to the application of phage T4 was reported. Phage T4 was detected in a dose-dependent manner in the feces of patients orally exposed to phages. However, oral phage application did not result in a decrease in total fecal *E. coli* counts. Moreover, no substantial replication of phage T4 was observed in the commensal population of *E. coli*.

Three of the four trials evaluating the clinical efficacy of phages showed negative results. As the other trials were designed to evaluate the safety, tolerability and/or preliminary efficacy of phage therapy, one cannot conclude on the clinical efficacy of phages based on their results.

### 3.2. Literature Reviews 

Twenty-one literature reviews evaluating the efficacy of phage therapy were identified. The following nine reviews covered the largest number of patients.

The 2021 literature review by Genevière et al. included 20 case reports of 51 patients treated with phages for bone and joint infections [28]. The overall success rate was 71%. 

In 2020, Clarke et al. published a literature review of 17 case reports assessing the safety and efficacy of phage therapy in 277 patients with bone and joint infections [33]. Clinical resolution was observed in 93.1% of cases.

The efficacy of phage therapy was evaluated in two meta-analyses: one by Abedon et al. (2011) and the other by Kutter et al. (2014) [53,57]. Both meta-analyses included studies carried out in the former Eastern Bloc. The disparity in methods used and the wide variety of treated infections make it difficult to conclude on the efficacy of evaluated treatments. 

In their 2012 literature review, Miecdzybrodzki et al. retrospectively evaluated 153 patients treated with phages at the Hirszfeld Institute of Immunology and Experimental Therapy in Poland between January 2008 and December 2010 [56]. They found phages to be well tolerated and to have overall efficacy.

In 2009, Chanishvili et al. reported several cases of patients treated with phages at the G. Eliava Institute in Georgia [58]. One cannot not conclude on the efficacy of phages due to the disparity in methods used and the range of treated infections. 

In their 2000 literature review, Weber-Dabrowska et al. from the Hirszfeld Institute of Immunology and Experimental Therapy in Poland reported 1307 cases of treatment with phages in patients with suppurative bacterial infections caused by different species of multidrug-resistant bacteria [64]. Their conclusions indicated a very high efficacy of phage therapy: complete cure was observed in 85.9% of cases, partial cure in 10.9% of cases, and failure in 3.8% of cases.

Two theses were identified that reviewed cases of patients treated with phages. The thesis by Domrault, published in 1998, reported 557 cases of treatment with phages in patients infected with resistant *P. aeruginosa*. Treatment was successful in the majority of cases [65]. In his 1931 thesis, the oldest to evaluate phage therapy, Pesce reviewed 622 cases of phage treatment and described his own use of phages in 14 patients with cutaneous infections. Here again, results were mainly positive [67]. 

The majority of reviews showed positive results. Seven reviews were not referenced in the MEDLINE database. Of these, two were found in articles (one in French and one in English), three in theses (all in French), and two in books (both in English). Thus, our literature review adds more than 1000 patients to the existing data.

### 3.3. Prospective Non-Randomized Clinical Trials 

Three prospective non-randomized clinical trials were identified.

The 2021 prospective study by Patel et al. evaluated the efficacy of phage therapy in 48 patients presenting with a wound that had failed to heal after 6 weeks of conventional therapy [32]. The patients received either a phage for single bacterial infection or a cocktail of phages specific to two or more infecting bacteria. Phage treatment was applied on the wound surface five to seven times over a period of 9 months, and patients were followed for 3 months. A cure rate of 81.2% was obtained.

In 2019, Gupta et al. conducted a prospective study in 20 Indian patients treated with phages for chronic nonhealing wounds infected with *E. coli*, *S. aureus,* and *P. aeruginosa* [41]. A cocktail of customized phages was applied over the wounds in three to five doses. A significant improvement in wound healing was observed, with seven patients cured at Day 21.

In their 1987 prospective study, Cislo et al. evaluated the effectiveness of phage therapy in 31 patients with chronic suppurative skin infections caused by *Pseudomonas* spp., *Staphylococcus* spp., *Klebsiella* spp., *Proteus* spp., and *Escherichia* spp. [66]. The outcome was favorable in 23 patients.

### 3.4. Isolated Case Reports

Our search of the literature identified 95 case reports published since 1945. These reports concerned more than 2500 patients, and the majority showed positive results (87/95). Phage therapy was associated with antibiotic therapy in 71 of the reports. A total of 12 case reports were not referenced in the MEDLINE database. Of these, eight were found in articles (four in French and four in English), and four in theses (all in French). These case reports add 200 patients to the existing data (Table 2). The main indications and the success rate associated were osteoarticular (94%, 16/17), pulmonary (86%, 12/14), skin (91%, 10/11), and digestive (77%, 7/9) infections.

A total of 119 case reports published before 1945 were identified. The majority showed positive results (101/118). These reports concerned nearly 4000 patients. A total of 102 case reports were not referenced in the MEDLINE database. Of the identified case reports, 78 were found in articles (45 in French and 33 in English), and 23 in theses (all in French). More than 2600 patients are thus added to the existing data (Table 3). The main indications and the success rate associated were bacteremia (79%, 15/19), skin (92%, 24/26), and digestive (86%, 12/14) infections.

### 3.5. Research Prospects

In 2017, Leitner et al. proposed a methodology for randomized controlled trials evaluating the use of phages in the treatment of urinary tract infections [282].

A 2020 literature review by Melo et al. reported the results of preclinical studies on phage treatments conducted in Western countries over the previous ten years [283]. Some of these results are encouraging and suggest the need to conduct randomized clinical trials to confirm the efficacy of phage treatments. 

## 4. Conclusions

This literature review identified multiple cases of successful treatment with phages in patients infected with different types of microorganisms. A total of 120 publications dealing with approximately 4000 patients were added to the existing data. However, the significant publication bias and the non-standardization of methods for the assessment of phage therapy do not allow us to conclude on the efficacy of phages. Relevant pharmacological data, including treatment dosage and duration, should be collected to help standardize evaluation methods. As we have shown previously, the literature also seems to show good results from the combination of phages and antibiotics, by a synergistic effect [284]. Prospective randomized clinical trials could then be conducted to confirm or disprove the clinical efficacy of different phage treatments. Lastly, literature reviews should be performed in different countries to identify non-referenced and/or non-translated publications on phage therapy.

## Figures and Tables

**Figure 1 antibiotics-12-00751-f001:**
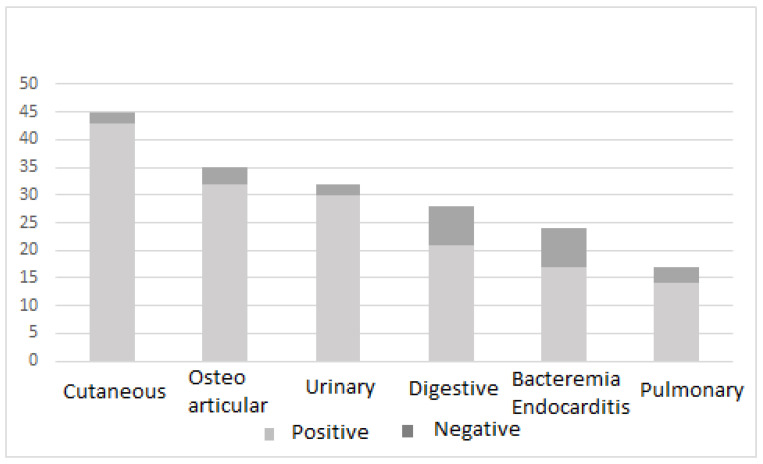
Distribution of articles on phage therapy for the main clinical foci of infection.

**Table 1 antibiotics-12-00751-t001:** Main prospective randomized clinical trials, literature reviews, and prospective non-randomized clinical trials.

Year	Number of Publications and/or Patients	Clinical Focus	Microorganism	Main Results	Reference
2022	41 reports/63 patients	Varied	Varied	Clinical success in 55 patients	[27]
2021	20 reports/51 patients	Osteoarticular	Varied	Clinical success	[28]
2021	9 reports/23 patients	Osteoarticular	Varied	Clinical success	[29]
2020	28 patients	Urinary	*P. aeruginosa*	Clinical failure	[30]
2021	14 clinical trials/35 case reports	Varied	Varied	Unspecified	[31]
2021	48 patients	Cutaneous	Varied	Clinical success	[32]
2020	277 patients	Osteoarticular	Varied	Clinical success	[33]
2020	13 studies	Diabetic foot infection	Varied	Clinical success in 11 studies Non-significant results in 2 studies	[34]
2020	13 patients	Severe infections	*S. aureus*	Safety demonstrated	[35]
2020	22 patients	Digestive	*E. coli*	Clinical success	[36]
2020	27 reports	Cutaneous	Varied	Clinical success	[37]
2019	10 studies	Varied	Varied	Clinical success in 5 studies; efficacy not determined in 5 studies	[38]
2019	36 patients	Digestive	*E. coli*	Safety demonstrated	[39]
2019	36 patients	Digestive	Varied	Safety demonstrated	[40]
2019	20 patients	Cutaneous	*E. coli* *P. aeruginosa* *S. aureus*	Clinical failure	[41]
2019	27 patients	Cutaneous	*P. aeruginosa*	Clinical failure	[42]
2019	12 studies	Varied	Varied	Clinical success	[43]
2019	8 studies	Varied	Varied	Clinical success	[44]
2019	29 case reports/1971 patients	Varied	Varied	Clinical success	[45]
2019	9 patients	ENT	*S. aureus*	Safety demonstrated	[46]
2019	6 studies	Varied	Varied	Safety demonstrated	[47]
2018	30 studies	Varied	*E. coli* *S. aureus* *Klebsiella pneumoniae* *Acinetobacter baumannii* *P. aeruginosa*	Clinical success	[48]
2018	10 patients	ENT	*S. aureus*	Safety demonstrated	[49]
2018	234 patients	Cutaneous	Unspecified	Clinical success	[50]
2016	No publications or patients added	Varied	Varied	Clinical success	[51]
2016	79 patients	Digestive	*E. coli*	Clinical failure	[52]
2014	No publications or patients added	Varied	Varied	Description of other studies	[53]
2013	No publications or patients added	Varied	*P. aeruginosa*	Clinical success	[54]
2012	No publications or patients added	Wounds	Varied	Description of other studies	[55]
2012	157 patients	Varied	Varied	Clinical success in 44% of patients	[56]
2011	No publications or patients added	Varied	Varied	Description of other studies	[57]
2009	No publications or patients added	Varied	Varied	Description of other studies	[58]
2009	39 patients	Cutaneous	*P. aeruginosa* *S. aureus* *E. coli*	Clinical failure	[59]
2009	24 patients	ENT	*P. aeruginosa*	Clinical success	[60]
2008	No publications or patients added	Varied	Varied	Description of other studies	[61]
2005	15 patients	Digestive	*E. coli*	Clinical failure	[62]
2001	No publications or patients added	Varied	Varied	Description of other studies	[63]
2000	1307 patients	Suppurative infections	Varied	Clinical success	[64]
1998	557 patients	Varied	*P. aeruginosa*	Clinical success	[65]
1987	31 patients	Cutaneous	*Staphylococcus* spp. *Pseudomonas* spp.	Clinical success	[66]
1931	636 patients	Cutaneous	*S. aureus*	Clinical success	[67]

**Table 2 antibiotics-12-00751-t002:** Case reports published since 1945.

Year	Number of Patients	Clinical Focus	Microorganism	Antibiotic Association Y/N	Main Results	Reference
2022	1	Cutaneous	*Mycobacterium chelonae*	Y	Clinical success	[68]
2021	1	Osteoarticular	*Staphylococcus epidermidis*	Y	Clinical success	[69]
2021	1	Osteoarticular	*P. aeruginosa*	Y	Clinical success	[70]
2021	1	Urinary	Varied	Y	Clinical success	[71]
2021	1	Osteoarticular	*P. aeruginosa*	Y	Clinical success	[72]
2021	1	Pulmonary	*Achromobacter* *xylosoxidans*	Y	Clinical success	[73]
2021	1	Digestive	*Enterococcus faecium*	Y	Clinical success	[74]
2021	1	Osteoarticular	*S. aureus*	Y	Clinical success	[75]
2021	1	Pulmonary	*A. baumannii*	Y	Clinical success	[76]
2021	4	Pulmonary	*A. baumannii*	Y	Clinical success in 2 patients; death in 2 patients	[77]
2020	10	Varied	Mainly, *P. aeruginosa* *S. aureus* *Mycobacterium abscessus*	Y	Clinical success	[78]
2020	1	Urinary	*K. pneumoniae*	Y	Clinical success	[79]
2020	1	Osteoarticular	*K. pneumoniae*	Y	Clinical success	[80]
2020	1	Osteoarticular	*S. aureus*	Y	Clinical success	[81]
2020	1	Cardiac	*S. aureus* and *Cutibacterium acnes*	Y	Clinical success	[82]
2020	3	Osteoarticular	*S. aureus*	Y	Clinical success	[83]
2020	1	Pulmonary	*Achromobacter* spp.	Y	Clinical success	[84]
2020	1	Cardiac	*S. aureus*	Y	Clinical success	[85]
2020	1	Urinary	*K. pneumoniae*	Y	Clinical success	[86]
2020	8	Cardiothoracic surgery	*S. aureus* *E. faecium* *P. aeruginosa* *K. pneumoniae* *E. coli*	Y	Clinical success	[87]
2019	1	Cardiac	*S. aureus*	Y	Clinical success	[88]
2019	3	Pulmonary	*P. aeruginosa* *Burkholderia dolosa*	Y	Clinical success	[89]
2019	1	Uro-digestive colonization	*K. pneumoniae*	Y	Clinical success	[90]
2019	1	Pulmonary	*M. abcessus*	Y	Clinical success	[91]
2019	13	Varied	Varied	Y	Clinical success	[92]
2019	15	Unspecified	Varied	Y	Unspecified	[93]
2019	1	Cardiac	*S. aureus*	Y	Clinical success	[94]
2019	1	Urinary	*K. pneumoniae*	Y	Clinical success	[95]
2019	1	Pulmonary	*P. aeruginosa*	Y	Clinical success	[96]
2019	1	Pulmonary	*P. aeruginosa*	Y	Clinical success	[97]
2019	1	Osteoarticular	*A. baumanii* and *K. pneumoniae*	Y	Clinical success	[98]
2019	4	Osteoarticular	*P. aeruginosa, S. epidermidis, S. aureus, Enterococcus faecalis*	Y	Clinical success	[99]
2019	1	Osteoarticular	*P. aeruginosa*	Y	Clinical success	[100]
2018	8	Bacteraemia, endocarditis and pulmonary	*S. aureus* and *P. aeruginosa*	Y	Clinical success	[101]
2018	1	Pulmonary	*P. aeruginosa*	Y	Clinical success	[102]
2018	1	Vascular prosthesis	*P. aeruginosa*	Y	Clinical success	[103]
2018	3	Osteoarticular	*S. aureus*	Y	Clinical success	[104]
2018	1	Osteoarticular	*P. aeruginosa*	Y	Local success but patient death	[105]
2018	6	Diabetic foot	*S. aureus*	Y	Clinical success	[106]
2018	1	Diabetic foot	*S. aureus*	Y	Clinical success	[107]
2018	1	Pulmonary	*A. xylosoxidans*	Y	Clinical success	[108]
2018	1	Neurosurgical	*A. baumanii*	N	Local success but patient death	[109]
2018	15	Varied	Mainly, *S. aureus*	Y	Clinical success in 12 patients	[110]
2018	9	Urinary	Varied	N	Clinical success	[111]
2017	1	Bacteraemia	*P. aeruginosa*	Y	Clinical success	[112]
2017	1	Urinary	*P. aeruginosa*	Y	Clinical success	[113]
2017	87	Cutaneous	*S. aureus*	N	Clinical success	[114]
2017	1	Digestive	*A. baumanii*	Y	Clinical success	[115]
2017	3	Cutaneous	*S. aureus*	N	Clinical success	[116]
2017	1	Cutaneous	*S. aureus*	N	Clinical success	[117]
2016	6	Diabetic foot	*S. aureus*	Y	Clinical success	[118]
2016	9	Urinary	Varied	N	Clinical success	[119]
2015	1	Ocular	*S. aureus*	Unknown	Clinical success	[120]
2014	9	Cutaneous	*S. aureus and P. aeruginosa*	N	Clinical failure	[121]
2011	1	Urinary	*P. aeruginosa*	Y	Clinical success	[122]
2011	1	Pulmonary	*S. aureus*	Y	Clinical success	[123]
2010	2	Osteoarticular	*S. aureus*	Y	Clinical success	[124]
2009	28	Ocular	*Staphylococcus* spp.	N	Clinical success	[125]
2009	3	Urinary	*E. faecalis*	N	Clinical success	[126]
2009	37	Osteoarticular	Mainly, *S. aureus*	Y	Significant improvement	[127]
2007	1	ENT	*S. aureus*	Y	Clinical success	[128]
2007	1	Pulmonary	*P. aeruginosa*	Y	Clinical success	[129]
2006	1	Digestive colonization	*S. aureus*	N	Clinical success	[130]
2006	1	Cutaneous	*P. aeruginosa*	Y	Clinical success	[131]
2006	1	ENT	*Staphylococcus hominis*	N	Clinical success	[132]
2005	2	Cutaneous	*S. aureus*	Y	Clinical success	[133]
2003	94	Sepsis	Varied	Y	Clinical success	[134]
2002	96	Cutaneous	Varied	N	Clinical success	[135]
2001	20	Varied	Varied	Unknown	Clinical success	[136]
1990	30	Cutaneous	*P. aeruginosa*	N	Clinical success	[137]
1987	550	Suppurative infections	Unspecified	Y	Clinical success	[138]
1985	273	Varied	Mainly *Staphylococcus* spp.	Y	Clinical success	[139]
1985	114	Varied	Varied	Y	Clinical success	[140]
1985	370	Varied	Varied	Y	Clinical success	[141]
1984	150	Varied	Varied	Y	Clinical success	[142]
1983	138	Varied	Varied	Y	Clinical success	[143]
1981	8	Cutaneous	*S. aureus*	N	Clinical success	[144]
1979	7	Osteoarticular	Varied	Y	Clinical success	[145]
1978	1	Vascular	*Serratia marcescens*	Y	Clinical failure	[146]
1971	25	Digestive	*Vibrio cholerae*	N	Clinical failure	[147]
1965	1	Digestive	*Salmonella panama*	N	Clinical success	[148]
1962	45	Digestive	*E. coli*	N	Prophylaxis; 1 clinical failure	[149]
1960	10	Varied	*P. aeruginosa*	Y	Clinical success	[150]
1959	34	Varied	Varied	Y	Clinical success	[151]
1959	1	Neurological	*P. aeruginosa*	Y	Clinical success	[152]
1958	1	Neurological	*E. coli*	Y	Clinical success	[153]
1953	2	Osteoarticular	*S. aureus* and *Streptococcus* spp.	N	Clinical success	[154]
1949	77	Varied	*Staphylococcus* spp.	N	Clinical success	[155]
1949	14	Digestive	Unspecified	N	Clinical success	[156]
1948	1	Osteoarticular	Unspecified	N	Clinical success	[157]
1947	1	Endocarditis	*Staphylococcus* spp.	Y	Clinical success	[158]
1946	56	Digestive	*S. typhi*	N	Unspecified	[159]
1946	9	Varied	*Staphylococcus* spp.	Y	Clinical success	[160]
1945	1	Endocarditis	*S. aureus*	Y	Clinical success	[161]
1945	7	Cutaneous	Mainly, *S. aureus*	N	Clinical success	[162]

ENT: ear-nose-throat; N: no; Y: yes.

**Table 3 antibiotics-12-00751-t003:** Case reports published before 1945.

Year	Number of Patients	Clinical Focus	Microorganism	Main Results	Reference
1943	45	Thrombophlebitis	*S. aureus*	Unspecified	[163]
1943	10	Neurological	*Staphylococcus* spp.	Clinical success	[164]
1942	385	Bacteraemia	*S. aureus*	Unspecified	[165]
1941	2	Neurological	*S. aureus*	No significant improvement	[166]
1941	5	Varied	*Staphylococcus* spp.	Clinical success	[167]
1941	1	Bacteraemia	*S. aureus*	Clinical success	[168]
1941	9	Surgical	Varied	Clinical success	[169]
1940	36	Bacteraemia	*S. aureus*	Clinical success	[170]
1940	12	Osteoarticular and bacteraemia	*S. aureus*	Clinical success	[171]
1938	8	Breast abscess	Mainly *Staphylococcus* spp.	Clinical success	[172]
1937	19	Urinary	*E. coli*	Clinical success	[173]
1937	88	Gynaecological	*Neisseria gonorrhoeae*	Clinical success	[174]
1937	5	Varied surgical	*S. aureus*	Clinical success	[175]
1937	3	Osteoarticular	*S. aureus*	Clinical success	[176]
1937	106	Osteoarticular and bacteraemia	*S. aureus*	Unspecified	[177]
1937	1	Osteoarticular	*S. aureus*	Clinical success	[178]
1937	2	Bacteraemia	*S. aureus*	Clinical success	[179]
1937	10	Cutaneous	Unspecified	Clinical success	[180]
1936	1	Bacteraemia	*S. aureus*	Clinical success	[181]
1936	27	Varied	*S. aureus*	Clinical success	[182]
1936	4	Neurological	*S. aureus*	Clinical success	[183]
1936	15	Bacteraemia	*S. aureus*	Clinical success	[184]
1936	100	Bacteraemia	*S. aureus*	Unspecified	[185]
1936	Around 10	Cutaneous	Unspecified	Clinical success	[186]
1936	1	Pulmonary	*E. coli*	Clinical success	[187]
1936	1	Pulmonary	Polymicrobial	Clinical success	[188]
1935	1	Bacteraemia	*S. aureus*	Clinical success	[189]
1935	1	Neurological	*S. aureus*	Clinical success	[190]
1935	2	Endocarditis	*S. aureus* and *Streptococcus viridans*	Clinical success	[191]
1935	>100	Cutaneous	Unspecified	Clinical success	[192]
1935	1	Cutaneous and osteoarticular	Unspecified	Clinical success	[193]
1935	1	Osteoarticular and bacteraemia	*Staphylococcus albus, Staphylococcus haemolyticus, Streptococcus* spp.	Clinical success	[194]
1935	Unspecified	Cutaneous	Unspecified	Unspecified	[195]
1935	1	Bacteraemia	*S. aureus*	Clinical success	[196]
1935	18	Cutaneous, urinary and ophthalmological	Varied	Clinical success	[197]
1935	34	Urinary	Mainly, *E. coli*	Clinical success	[198]
1934	4	Varied	Varied	Clinical success	[199]
1934	14	Urinary	*Colon bacillus*	Clinical success	[200]
1934	4	Bacteraemia	*Colon bacillus*	No significant improvement	[201]
1934	14	Digestive	Unspecified	Clinical success	[202]
1934	1	Osteoarticular	*Streptococcus* spp. and *Staphylococcus* spp.	Clinical success	[203]
1934	1	Bacteraemia	*S. aureus*	Clinical success	[204]
1934	2	Digestive	Unspecified	Clinical success	[205]
1934	7	Cutaneous	*Staphylococcus* spp.	Clinical success	[206]
1934	1	Osteo-articular	*Streptococcus* spp.	Clinical success	[207]
1934	110	Cutaneous	*Staphylococcus* spp.	Clinical success	[208]
1933	100	Osteoarticular	Mainly *Staphylococcus* spp.	Significant improvement	[209]
1933	22	Surgical	Unspecified	Unspecified	[210]
1933	200	Cutaneous	Unspecified	Unspecified	[211]
1933	8	Digestive	*Salmonella typhi* and *S. paratyphi*	Clinical success	[212]
1933	9	Cutaneous	*Staphylococcus* spp.	Clinical success	[213]
1933	1	ENT	*Staphylococcus* spp.	Clinical success	[214]
1933	49	Gynaecological and bacteraemia	Varied	Some clinical success	[215]
1933	1	Osteoarticular	*Staphylococcus* spp.–*S. viridans*	Clinical success	[216]
1933	1	Osteoarticular	*Staphylococcus* spp.	Clinical success	[217]
1933	2	ENT	*S. aureus*	Clinical success	[218]
1933	30	Surgical	Varied	Clinical success	[219]
1933	1	Bacteraemia	Unspecified	Clinical success	[220]
1932	10	Osteoarticular	*S. aureus*	Clinical failure	[221]
1932	4	Urinary	*E. coli*	Clinical success	[222]
1932	8	Digestive	*Dysenteric bacillus*	Clinical success	[223]
1932	173	Plague	Yersin’s bacillus	Clinical success	[224]
1932	1	Digestive	*Dysenteric bacillus*	Clinical success	[225]
1932	7	Cutaneous and bacteraemia	*Staphylococcus* spp. and *E. coli*	Clinical success	[226]
1932	7	Cutaneous, pulmonary and urinary	Varied	Clinical success	[227]
1932	1	Neurological	*S. aureus*	Clinical success	[228]
1932	7	Urinary and gynaecological	Varied	Clinical success	[229]
1932	191	Urinary	*E. coli*	Clinical failure	[230]
1931	266	Digestive	*V. cholerae*	Clinical success	[231]
1931	4	Surgical	Varied	Clinical success	[232]
1931	Unspecified	Varied	Varied	Unspecified	[233]
1931	1	Pulmonary	Unspecified	Clinical success	[234]
1931	46	Urinary	Mainly, *E. coli*	Clinical success	[235]
1931	14	ENT	Unspecified	Clinical success	[236]
1931	20	Cutaneous	*Staphylococcus* spp.	Clinical success	[237]
1931	1	Digestive	Unspecified	Clinical success	[238]
1931	1	Osteoarticular	Unspecified	Clinical success	[239]
1931	115	Cutaneous	*Staphylococcus* spp.	Clinical success	[240]
1931	1	Osteoarticular	*S. aureus*	Clinical success	[241]
1931	1	Cutaneous	Unspecified	Clinical success	[242]
1931	2	Cavernous sinus thrombophlebitis	*S. aureus*	No significant improvement	[243]
1931	1	Osteoarticular	*Staphylococcus* spp.	Clinical success	[244]
1930	6	Cutaneous	Ducrey’s bacillus	Clinical success	[245]
1930	13	Cutaneous	Mainly, *S. aureus*	Clinical success	[246]
1930	4	Digestive	*S. typhi*	Clinical success	[247]
1930	5	Urinary	*E. coli*	Clinical success	[248]
1930	2	Cutaneous	Unspecified	Clinical success	[249]
1930	1	Bacteraemia/Urinary	*E. coli*	Clinical success via intravenous route; clinical failure via intravesical route and subcutaneous	[250]
1929	1	Cutaneous	*S. aureus*	Clinical success	[251]
1929	66	Digestive	*Dysenteric bacillus*	Clinical success	[252]
1929	1	Bacteraemia	*S. aureus*	Clinical success	[253]
1929	264	Cutaneous	*S. aureus*	Clinical success	[254]
1929	1	ENT	Unspecified	Clinical success	[255]
1929	1	ENT	Unspecified	Clinical success	[256]
1929	3	ENT	Varied	Clinical success	[257]
1929	>300	Varied	Varied	Clinical success	[258]
1929	76	Varied	*S. aureus* *E. coli*	Clinical success	[259]
1928	12	Urinary	*E. coli* and *P. aeruginosa*	Clinical success	[260]
1928	65	Urinary, digestive and other	Varied	Clinical failure	[261]
1928	Around 200	Varied	*S. aureus* *S. albus* *E. coli* *P. aeruginosa*	Clinical success	[262]
1927	3	Digestive	*D. bacillus*	Clinical success	[263]
1926	58	ENT	Mainly, *S. aureus*	Clinical success	[264]
1926	>100	Cutaneous, ENT and urinary	*S. aureus*	Clinical success	[265]
1926	4	Urinary	*E. coli*	Clinical success	[266]
1925	30	Urinary	*E. coli* and *S. aureus*	Clinical success	[267]
1925	4	Bubonic plague	*Yersinia pestis*	Clinical success	[268]
1925	7	Urinary	*E. coli*	Clinical success	[269]
1924	11	Urinary and digestive	*E. coli* *Salmonella* spp.	Clinical success	[270]
1924	4	Cutaneous	*S. aureus*	Clinical success	[271]
1924	14	Cutaneous	*S. aureus*	Clinical success	[272]
1923	11	Urinary	*E. coli*	Clinical success	[273]
1923	2	Cutaneous	*S. aureus*	Clinical success	[274]
1923	1	Digestive	*Salmonella* spp.	Clinical success	[275]
1923	7	Varied	Varied	Clinical success	[276]
1923	1	Urinary	*E. coli*	Clinical success	[277]
1922	7	Cutaneous	*S. aureus*	Clinical failure	[278]
1922	6	Cutaneous	*S. aureus*	Clinical success	[279]
1922	1	Urinary	*E. coli*	Clinical success	[280]
1922	12	Digestive	*E. coli*	No significant improvement	[281]

ENT: ear-nose-throat.

## Data Availability

Not applicable.

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
