# Peer review of "A Century of Clinical Use of Phages: A Literature Review"

_antibiotics, 2023, doi:10.3390/antibiotics12040751_

Round 1

Reviewer 1 Report

Dear respected authors, 

I suggest some comments that may improve your work. 

1-The authors concluded their results based only on the MEDLINE database. I suggest using additional databases like Embase to ensure they found all relevant results for their search.

2-The authors mentioned the usage of phages in food processing and plant disease control, while the review focused only on phage therapy in humans and there is no relevant data regarding food and plant in the result section.

3-The exclusion criteria should be mentioned in the method section.

4-I suggest if possible, adding a schematic diagram to conclude the results.

5-Table 3 which shows the case reports published before 1945, I think it is better to be supplementary data.

Author Response

Dear respected authors,

I suggest some comments that may improve your work.

Dear reviewer, we thank you for your attentive reading, and your comments which improve the quality of our manuscript.

1-The authors concluded their results based only on the MEDLINE database. I suggest using additional databases like Embase to ensure they found all relevant results for their search.

Unfortunately, we did not have the time due to the time allowed for the correction to rigorously analyze all the articles selected by other databases in order to ensure the completeness of the search on these bases. However, our search was expanded in addition to Medline for non-Medline referenced articles found in the references of read articles.

2-The authors mentioned the usage of phages in food processing and plant disease control, while the review focused only on phage therapy in humans and there is no relevant data regarding food and plant in the result section.

We proposed to describe the use of phage therapy in the food industry because it has been approved by health authorities, and its impact on the consumption of products by humans was considered negligible. We wanted by this paragraph to emphasize the harmlessness of the phages used and to support the fact that the health authorities already endorse their indirect use in humans.

3-The exclusion criteria should be mentioned in the method section.

Page 4, we added the following sentence: “We excluded articles that did not deal with the use of phages in humans and those referring to articles already cited that did not bring new cases to our analysis.”

4-I suggest if possible, adding a schematic diagram to conclude the results.

Page 4, we added the following sentence:  “Figure 1 shows the distribution of articles on phage therapy for the main clinical foci of infection.”

Figure 1 has been added page 14.

5-Table 3 which shows the case reports published before 1945, I think it is better to be supplementary data.

We wanted to show all the cases described for a century to support the harmlessness of this treatment and highlight the success rate described. We have arbitrarily chosen to put the limit between tables 2 and 3 at 1945. As the limit is arbitrary, we think that there would be less interest in showing only part of the results. If the length of the tables is too long, we could discuss with the editor to put tables 2 and 3 in supplementary data.

Reviewer 2 Report

This is nice compilation of the outcome of phage theapy efforts over the last 100 years. Very valuable for the reader is the huge collection of referenced studies and single cases in the tables. The only thing I am missing is in chapter 4 a more detailed judgement on the efficiacy of phage therapy in terms of the clinical focus, since this is a major difference in treatment response. In addition, a conclusive remark on the relevance of a combined treatment of phage and antibiotics would be valuable. I recommend the authors to add a respective paragraph on these points.

Minor remarks:

lines 29, 98: in vivo and in vitro should be in italic letters  

line 33: The authors should think about including a citation on the most recent classifications of the ICTV committee

line 41: please mention here that phage preparations have to be very clean and endotoxin free

line 101: spell out GMP

line 203: acute bacterial infection or diarrhoe (52).

Author Response

Dear reviewer, we thank you for your attentive reading, and your comments which improve the quality of our manuscript.

This is nice compilation of the outcome of phage theapy efforts over the last 100 years. Very valuable for the reader is the huge collection of referenced studies and single cases in the tables. The only thing I am missing is in chapter 4 a more detailed judgement on the efficiacy of phage therapy in terms of the clinical focus, since this is a major difference in treatment response. In addition, a conclusive remark on the relevance of a combined treatment of phage and antibiotics would be valuable. I recommend the authors to add a respective paragraph on these points.

On page 9, we added the following sentence: “The main indications and the success rate associated were osteoarticular (94%, 16/17), pulmonary (86%, 12/14), skin (91%, 10/11) and digestive (77%, 7/9) infections.”.

On page 11, we added the following sentence: “The main indications and the success rate associated were bacteraemia (79%, 15/19), skin (92%, 24/26) and digestive (86%, 12/14) infections.”

On page 15, we added the sentence: “As we have shown previously, the literature also seems to show good results from the combination of phages and antibiotics, by a synergistic effect [285]”.

Minor remarks:

lines 29, 98: in vivo and in vitro should be in italic letters  

We have made the requested changes.

line 33: The authors should think about including a citation on the most recent classifications of the ICTV committee

We have updated the reference as requested.

line 41: please mention here that phage preparations have to be very clean and endotoxin free

We have added the requested element.

line 101: spell out GMP

We spelled the abbreviation.

line 203: acute bacterial infection or diarrhoe (52).

We have made the requested changes.